# Geochemical Indicators for Paleolimnological Studies of the Anthropogenic Influence on the Environment of the Russian Federation: A Review

**Zakhar Slukovskii** [1,2]

1 Institute of the North Industrial Ecology Problems of Kola Science Center of RAS, 184209 Apatity, Russia; z.slukovskiy@ksc.ru or slukovsky87@gmail.com

2 Institute of Geology of Karelian Research Centre of RAS, 185910 Petrozavodsk, Russia

**Abstract:** Lake sediments are a reliable source of information about the past, including data of the origin of water bodies and their changes. Russia has more than 2 million lakes, so paleolimnological studies are relevant here. This review deals with the most significant studies of sequential accumulation of pollutants, including heavy metals in recent lake sediments in Russia. The key areas are northwestern regions of Russia (Murmansk Region, the Republic of Karelia, Arkhangelsk Region), the Urals (Chelyabinsk Region, the Republic of Bashkortostan), and Siberia. The review presents the data of pollutants accumulation, the sedimentation rate in lakes in the anthropogenic period, and the key sources of pollution of the environment in each of the mentioned regions. The article is divided into three parts (sections): industrial areas, urbanized areas, and background (pristine) areas so that readers might better understand the specifics of particular pollution and its impact on lake ecosystems. The impact of metallurgical plants, mining companies, boiler rooms, coal and mazut thermal power plants, transport, and other anthropogenic sources influencing geochemical characteristics of lakes located nearby or at a distance to these sources of pollution are considered. For instance, the direct influence of factories and transport was noted in the study of lake sediments in industrial regions and cities. In the background territories, the influence of long-range transport of pollutants was mainly noted. It was found that sedimentation rates are significantly lower in pristine areas, especially in the Frigid zone, compared to urbanized areas and industrial territories. In addition, the excess concentrations of heavy metals over the background are higher in the sediments of lakes that are directly affected by the source of pollution. At the end of the article, further prospects of the development of paleolimnological studies in Russia are discussed in the context of the continuing anthropogenic impact on the environment.

**Keywords:** freshwater ecosystems; lake sediments; human impact; heavy metals; Russia; Arctic

## 1. Introduction

The anthropogenic impact on the environment for the last two or three centuries is an indisputable fact. One of the well-known manifestations of this process is chemical pollution, reflected in the increased concentrations of various elements and substances in the main environmental components—air, soil, surface and underground waters, sediments of water bodies, and living organisms. Heavy metals and metalloids are among the most dangerous environmental pollutants, as their compounds are quite stable, can exhibit toxic properties, migrate along trophic chains from abiotic components of ecosystems to biota, and accumulate in sediments, soils, tissues, and organs of organisms [1–4]. Paleo-archive methods allow for simultaneous analysis of the current state of the environment and the historical trends often under the influence of the anthropogenic factors. Environmental archives often examined for anthropogenic contamination include ice and

tree cores, and peat and lake sediments cores [5–9]. Lake sediments can best perform the present and past environmental assessments of anthropogenic metal contamination, as lakes sediments act as a passive sampler of the environment, can be readily dated with radiometric methods, and are generally common in the vicinity of urban and industrial centers.

All over the world, researchers conduct paleolimnological reconstructions based on the detailed (layer-by-layer) study of sediment cores of the lakes, thus restoring the main stages of the anthropogenic influence on the studied water bodies and their surrounding areas [5,10–14]. Such works are certainly widely developed in Russia, where there are more than 2 million lakes with a surface area of ~350 thousand km² (excluding the Caspian sea). Paleolimnological studies and reconstructions are especially relevant for regions with large industrial histories (the Southern Ural, Murmansk Region, Western Siberia) [15–17]. Besides, the close location of the aquatic ecosystems to the direct sources of the anthropogenic emissions is important for such research. Therefore, paleolimnological studies are either impossible or barely conducted in the regions with a small number of lakes or in inaccessibility areas.

This review aims to highlight the main paleolimnological studies of the anthropogenic impact on the environment of the Russian Federation, published so far in Russian (in most cases) and English in scientific journals, books, and theses. This is extremely important, since, for example, this review of studies of natural archives [18] is not full without the data of Russian scientists. The focus was on the regions of Russia, where paleolimnological studies based on the analysis of the accumulation dynamics of heavy metals and metalloids in recent sediments have long been a part of the environmental monitoring system. These are the regions of Northwest Russia, the Southern Ural, and Siberia (Figure 1).

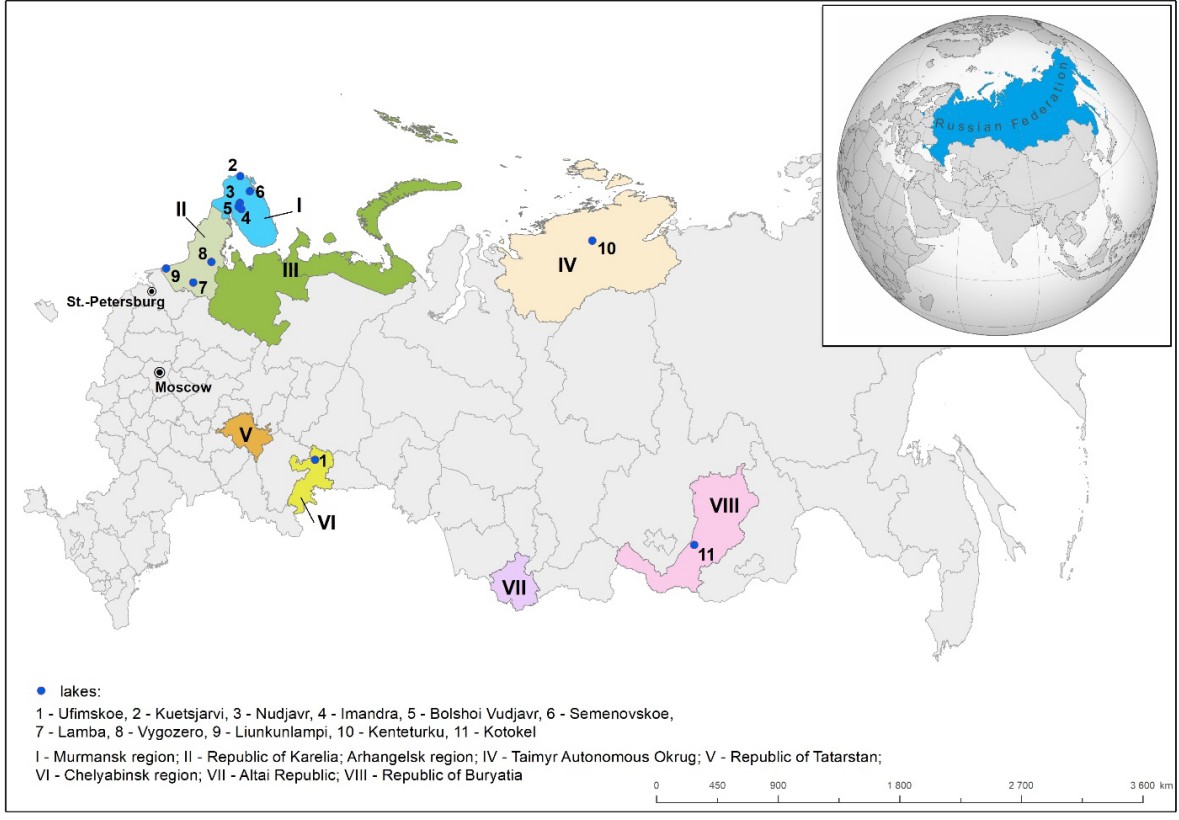

**Figure 1.** A map with the designation of lakes and key regions of paleolimnological studies. The main characteristics of water bodies from the map are in Table 1.

**Table 1.** Main parameters of water bodies are shown on the map in Figure 1. Note: n/d—there are no data.

| Water Bodies | Area | Coordinates | Square, km² | Depth, m | |
|---|---|---|---|---|---|
| | | | | Maximum | Average |
| Lake Ufimskoe | Chelyabinsk Region | 60.11862, 55.52231 | 0.89 | 3.5 | 1.1 |
| Lake Kuetsjarvi | | 30.16771, 69.43524 | 17.00 | 37.0 | n/d |
| Lake Nudjavr | | 32.88535, 67.92346 | 3.97 | 2.0 | 1.6 |
| Lake Imandra | Murmansk Region | 33.08029, 67.64688 | 876.00 | 67 | 13.3 |
| Lake Bolshoi Vudjavr | | 33.67456, 67.63246 | 3.49 | 38.6 | n/d |
| Lake Semenovskoe | | 33.09001, 68.99101 | 0.21 | 11.3 | 2.4 |
| Lake Lamba | | 34.24950, 61.80713 | 0.01 | 5.2 | 3.4 |
| Vygozero Reservoir | Republic of Karelia | 34.69694, 63.59750 | 1270 | 25 | 7.1 |
| Lake Liunkunlampi | | 29.87730, 61.49913 | 0.1 | 6.8 | 3.6 |
| Lake Kenteturku | Krasnoyarsk region | 96.43925, 73.46444 | 2.5 | 20 | 10 |
| Lake Kotokel | Republic of Buryatia | 108.15000, 52.81667 | 70 | 14 | 4.25 |

## 2. Materials and Methods

Publications, including articles, conference materials, books, and chapters in books published so far to the end of 2021, studied by the author, were used to prepare the review. The main criterion for using publications was the presence of the data on the studies of cores (up to 1 m) sediments of lakes and water bodies with the analysis of the layer-by-layer distribution of chemical elements (mainly heavy metals) and/or isotopes of $^{210}$Pb or $^{137}$Cs in these cores. Although the most studied materials were published in Russian language, they are still important for world science, as researchers have been using methods recognized in paleolimnology for studying geochemistry and the age of sediments of water bodies. This review will allow scientists from all over the world who do not speak Russian to become better acquainted with these studies, considering that Russia is a country with one of the largest number of lakes in the world, and thus has some of the largest numbers of limnological studies which should be known and recognizable. All the publications in Russian are marked in References as (in Russian).

Another criterion for choosing publications was dividing recent sediments cores by researchers into layers no more than 5 cm, with a few exceptions of 10 cm. Personal experience shows that larger layers do not allow for accurately assessing the impact of sources of anthropogenic emissions on the aquatic ecosystem. The best option is to divide cores into 1–2 cm layers, however, studies where cores were divided into 3–10 cm layers were also included in the review. Besides, the review focused on studying lake ecosystems, with rare exception being reservoir ecosystems. This choice resulted from the fact that the water bodies with relatively stagnant water are best suitable for paleolimnological reconstructions as sedimentary material does not mix, and thus accumulates more sequentially, which allows for accurately fixing various changes in the water body and its catchment area. River sediments were excluded from the review as sedimentary material in rivers

accumulates in a dynamic environment constantly mixing, which can provide only a general picture of sediment geochemistry. Marine sediments were also excluded since the sedimentation rate in seas, oceans, and large marine water bodies at all is usually very low, which does not allow for fixing point changes in the geochemistry of recent sediments over the last 100–300 years.

Concentrations of chemical elements in the article are presented in mg/kg. If concentrations could be taken only from charts, graphs, or figures, then approximate concentrations were used. All figures in this review are made by the author, and are made either on the basis of the numerical data from publications or the charts from the same works. In this case, there is no copyright infringement as charts were not copied—they were taken from open access sources and then remade either to another format or using other software for illustrations. In special cases, the researchers gave permission to use their data.

## 3. Results and Discussion

### 3.1. Industrial Areas

3.1.1. Ural Region

Chelyabinsk and Murmansk Regions are some of the most industrially developed regions of Russia. There are metallurgical plants for mining and processing copper and copper–nickel ores in both regions. There are also a large number of lakes subject to substantial pollution due to operations of these industrial enterprises in these regions [19–21]. Karabashmed (Karabashshkiy Copper-Smelting Plant) (the city of Karabash, Figure 2), producing blister copper, has been operating in Chelyabinsk Region since 1910. Many paleolimnological studies assessing the dynamics of pollutants in water bodies of the Chelyabinsk Region have been conducted in the impact area of this plant. For instance, the analysis of dynamics of heavy metals and stable $^{210}$Pb isotopes behavior in the core of recent sediments of Lake Serebry located 4 km from Karabashmed showed increased concentrations of Cu (up to ~6000 mg/kg, while background level is about 50 mg/kg), Zn (up to ~6000 mg/kg, background is ~70 mg/kg), Pb (up to ~2000 mg/kg, background is ~20 mg/kg), and Mn (up to ~1000 mg/kg, background is ~410 mg/kg) in the upper layers compared to the lower ones [16,22,23]. The increase in concentrations of these metals started according to different references at a depth of 50–80 cm, likely corresponding to the start of the plant operations. The average sedimentary rate in Lake Serebry in the industrial period was 4.8 mm/year [22]. However, more recent data show that this value can be higher, up to ~9 mm/year (calculated based on data from [16]).

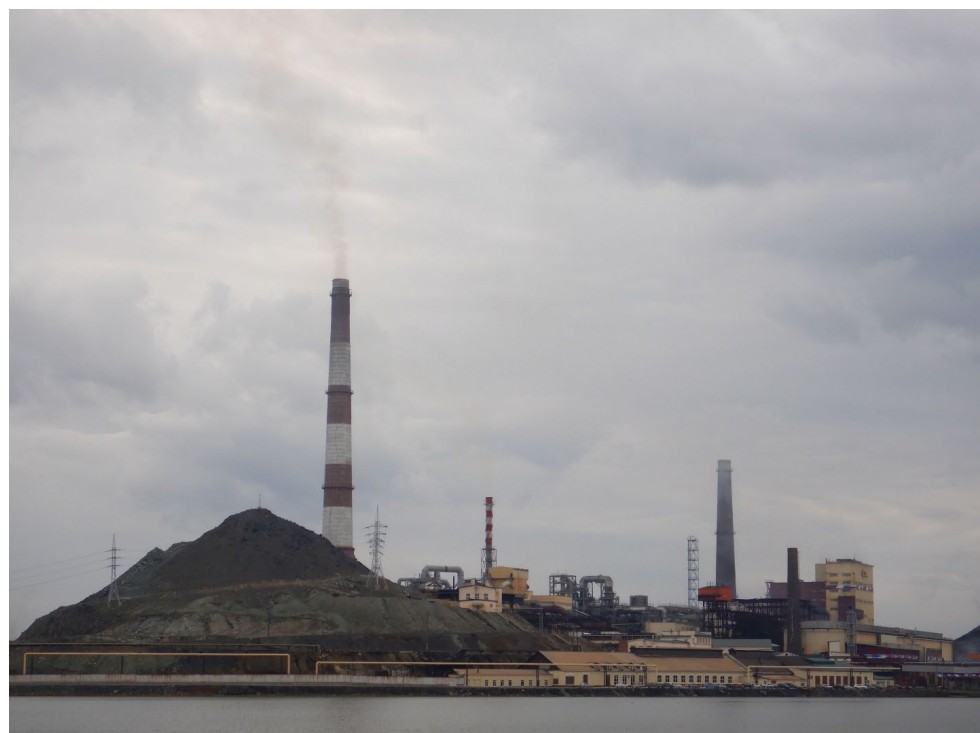

**Figure 2.** The view of Karabashskiy Copper-Smelting Plant (photo by the author).

Similar trends of heavy metals (Cu, Zn, Pb, Sb, Cd) can also be seen in sediments of other lakes located in the impact area of Karabashskiy Copper-Smelting Plant [19,23]. For instance, this is well-demonstrated in the example of Lake Ufimskoe located 7 km from the plant (Figure 3). The uppermost layers of lake sediments are enriched with Cu (up to 2341 mg/kg while the background level is ~120 mg/kg), Zn (up to 1256 mg/kg, background is ~54), Pb (up to 1039 mg/kg, background is ~8), Sb (up to 21 mg/kg, background is ~0.3), and Cd (up to 13 mg/kg, background is ~0.4) [19]. These metals are closely related to the copper-smelting plant operations. Other elements (e.g., V, Co, Li, rare earth elements, etc.) did not have a similar tendency towards an increase in the upper layers compared to the lower ones, which indicates that their origin in sediments is not related to the anthropogenic impact on the water body [19].

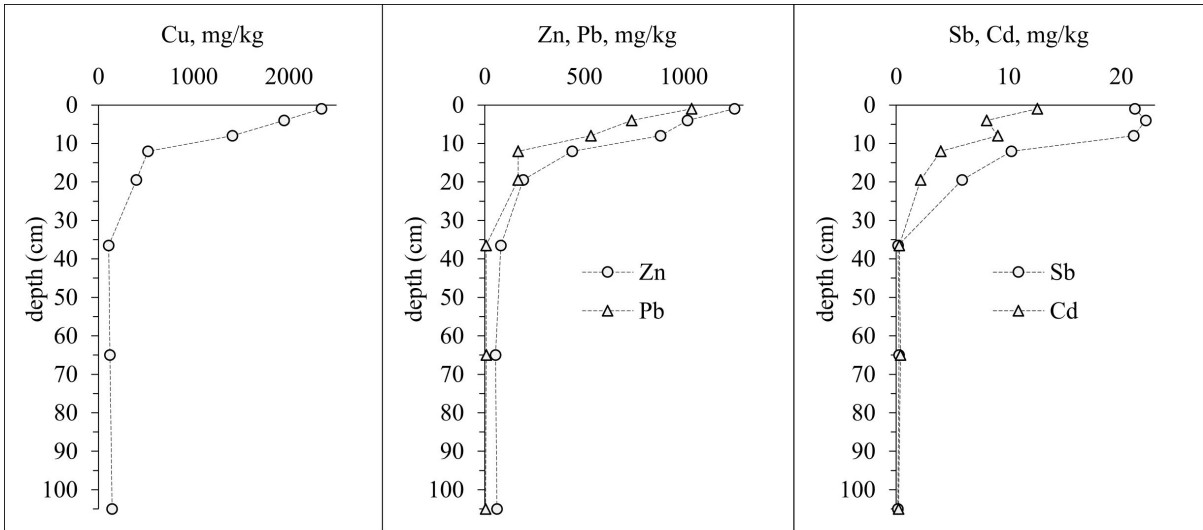

**Figure 3.** The vertical distribution of heavy metals in sediments of Lake Ufimskoe (Chelyabinsk Region) [19].

Scientists from the Institute of Mineralogy, Ural Branch of the Russian Academy of Sciences, revealed the similar dynamics of accumulation of heavy metals in Lake Syrytkul (~30 km from the plant) and Lake Turgoyak (~40 km from the plant) [22,23]. However, total values of Cu, Zn, and Pb concentrations in the upper layers of sediments of these lakes were lower than in Lake Serebry and Lake Ufimskoe. It was noted that the concentration of Cu reached 800 mg/kg, Zn—260 mg/kg, Pb—200 mg/kg in sediments of Lake Turgoyak, where the sedimentation rate was 1.7 mm/year [22]. Thus, there is a tendency towards decreasing concentrations of heavy metals in lake sediments with increasing distance to Karabashskiy Copper-Smelting Plant. In the north of Chelyabinsk Region, 100 km from the city of Karabash, Cu concentrations in recent sediments (0–16 cm) of Lake Itkul varied from 58 to 91 mg/kg, Zn from 94 to 228 mg/kg, and Pb from 20 to 64 mg/kg [19]. However, researchers include these three elements and Cd, Bi, Sb, Co, and Te in the anthropogenic geochemical association of studied sediments of Lake Itkul, as there is a stable tendency towards their increased concentrations in upper layers compared to the background level.

Furthermore, in the Urals (the Republic of Bashkortostan), Cu, Zn, Co, and Ni were also studied in recent sediments of Lake Bolshye Uchaly subject to the Uchaly geotechnical system (the city of Uchaly) [24]. Paleolimnological studies indicated that due to massive quarry blasting in the 1970–1980s and aerial dust from the processing plant, upper layers of sediments aged 40 years of Lake Bolshye Uchaly were enriched with Zn (up to ~6000 mg/kg, while the background level is ~100 mg/kg), Cu (up to ~600 mg/kg, background is ~200 mg/kg), Ni (up to ~45 mg/kg, background is ~27 mg/kg), Co (up to ~15 mg/kg, background is ~10 mg/kg), and Cd (up to ~4 mg/kg). Scientists associate these processes not only with the mine and the concentrating plant operations but also with transport emissions, which is reflected in increased concentrations of Pb (up to ~70 mg/kg, while the background level of Pb in lake sediments of Ural region is 21 mg/kg) in sediments of this water body [24].

### 3.1.2. Murmansk Region

The main sources of pollution in Murmansk Region (Figure 4) are two plants of Kola Mining and Metallurgical Company (Kola MMC), located near the city of Monchegorsk–"Severonickel" combine (the central part of the region) and the urban-type settlement of Nikel–"Pechenganickel" combine (the northwestern part of the region, near the Norway–Russia border) [20]. As the company deals with the mining and processing of copper–nickel ore, the key pollutants of lakes nearby are heavy metals Ni and Cu. Both combines started operating in the 1930s, which caused a significant anthropogenic load on terrestrial and aquatic ecosystems nearby [25–27].

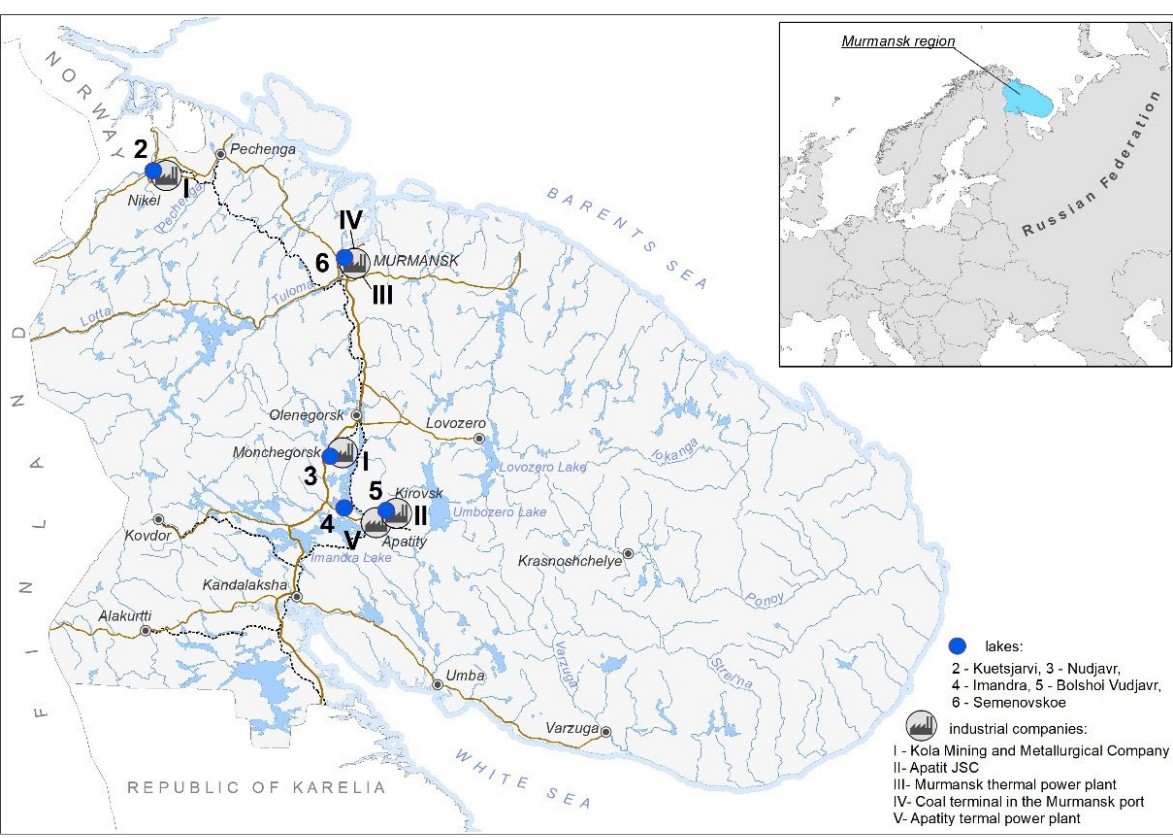

**Figure 4.** A map of Murmansk region with key lakes, cities, and industrial companies that are noted in the article.

In the area of Nikel, paleolimnological studies of the anthropogenic load on the aquatic ecosystems were mostly focused on lakes of the Pasvik river system. The largest lake, on the banks of which the plant operates (Figure 5), is Lake Kuetsjarvi. Researchers in Institute of the North Industrial Ecology Problems of Kola Science Center of Russian Academy of Sciences have been conducting environmental monitoring of this water body for about 30 years [27]. Studies showed that the upper 10–15 cm of sediments of Lake Kuetsjarvi were polluted with heavy metals [28,29]. The sedimentation rate in the lake varied from 1.5 to 3 mm/year depending on the study area [29]. The increases in concentrations of Ni (up to 4892 mg/kg, while background level is 32 mg/kg), Cu (up to 1496 mg/kg, background is 40 mg/kg), Zn (up to 301 mg/kg, background is 80 mg/kg), Co (up to 184 mg/kg, background is 16 mg/kg), Cd (up to 3.14 mg/kg, background is 0.10 mg/kg), Pb (up to 45.7 mg/kg, background is 6.6 mg/kg), As (up to 59.3 mg/kg, background is 2.6 mg/kg), and Hg (0.57 mg/kg, background is 0.05 mg/kg) were noted at all studied sites of Lake Kuetsjarvi (Figure 6), which is related to the start of the metallurgical plant operations in the 1930s. In 2020, the melting shop stopped working, which will probably lead to a decrease in the anthropogenic load on the lake. However, due to the pollution of soils around the water body with heavy metals, pollutants will continue to enter the Kuetsjarvi.

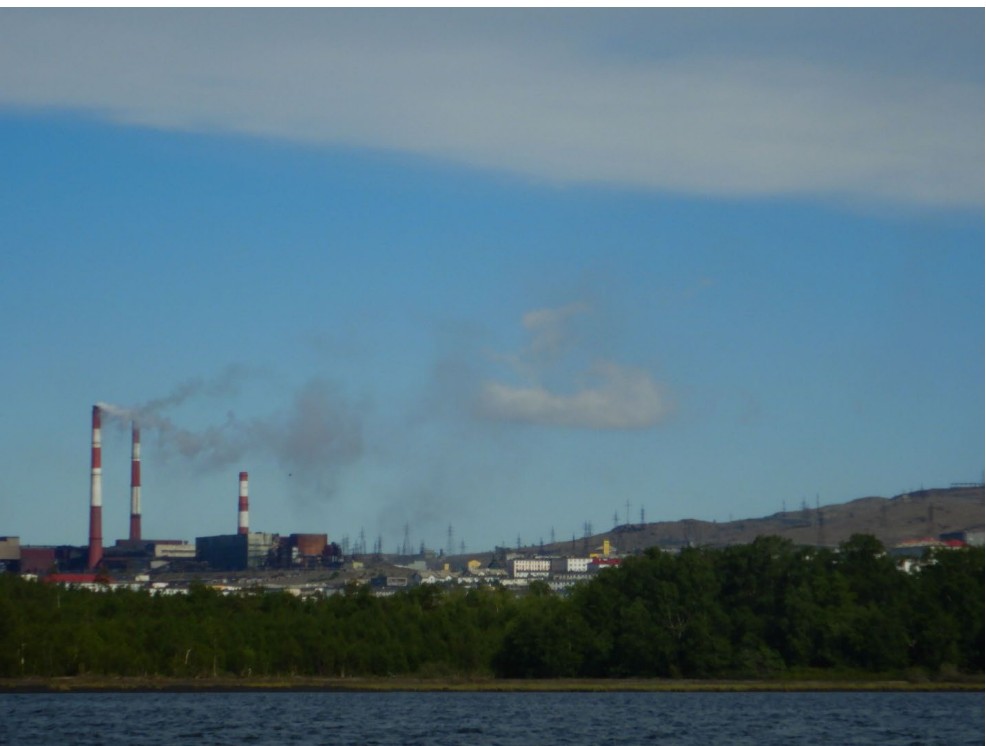

**Figure 5.** The view of Kola Mining and Metallurgical Company, "Pechenganickel" combine (photo by the author). Lake Kuetsjarvi is in the foreground.

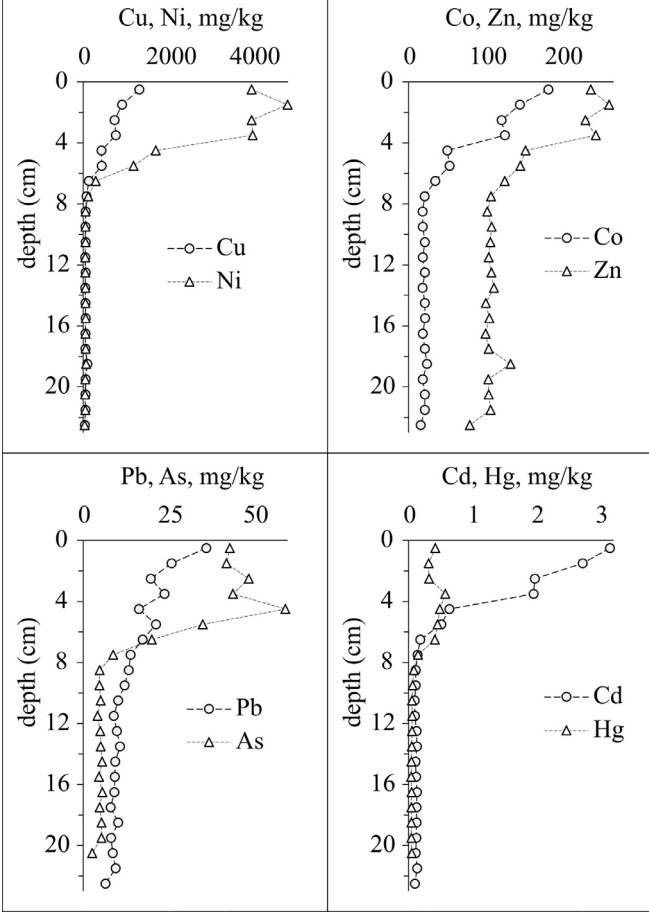

**Figure 6.** The vertical distribution of heavy metals in sediments of Lake Kuetsjarvi (Murmansk Region) [28,29].

Similar tendencies of accumulation of heavy metals in upper layers of sediments were also noted in other lakes located in the northwestern part of Murmansk Region and in the lakes of the north of Finland and Norway, also subject to the influence of Kola MMC ("Pechenganickel" combine). For instance, increased concentrations of Ni (up to 373 mg/kg, while background level is 45 mg/kg), Cu (up to 185 mg/kg, background is 38 mg/kg), and Co (up to 35 mg/kg, background is 21 mg/kg) were found in the upper layer of sediments of Lake Bjørnevatn located ~10 km to the north from the metallurgical plant [30,31]. Only the uppermost 4 cm of sediments were polluted with both heavy metals, which indicated the low sedimentation rate in this water body (~1 mm/year). In total, based on the data on the distribution of radionuclides [137]Cs and [210]Pb in sediment cores in the area of borders of Russia, Finland, and Norway, the average sedimentation rates varied from 0.65 to 3 mm/year [30,32]. In the sediment core of Lake Rabbvatnet (Norway) located ~40 km to the north from the metallurgical plant Ni concentrations reached ~250 mg/kg, Cu—~300 mg/kg, and Co—~12 mg/kg despite the distance [32]. As in other studied lakes, the increases in the content of mentioned heavy metals in sediments of this lake were found in layers dating to the 1920–1930s, and maximum concentrations were fixed in the 1970–1980s due to the most intensive work of the plant and the highest atmospheric emissions of pollutants. The impact of Kola MMC was also noted in the lakes of the north of Finland [33]. For instance, a slight increase in the concentrations of Ni (from ~14 to ~20 mg/kg) and Cu (from ~18 to ~23 mg/kg) was fixed in the uppermost 2–3 cm of sediments of Lake Vassikajarvi, despite the fact that this water body is located ~150 km from the direct source of pollution. Therefore, similarly to research in Chelyabinsk Region, limnological studies in the northwest of Murmansk Region and in the border area showed a tendency towards decreasing concentrations of main pollutants from the metallurgical plant in the upper layers of lake sediments with increasing distance from the industrial enterprise. The negative impact of emissions from the second combine of Kola MMC located near the city of Monchegorsk was also well studied on the example of lakes, including Lake Imandra, which is the largest water body in Murmansk Region [34,35]. The metallurgical plant in Monchegorsk ("Severonickel" combine) started operating in the 1930s refining copper–nickel ore. By mass, these elements (Ni and Cu) are the main pollutants of the local environment. One of the most polluted water bodies of this region is Lake Nudjavr, receiving waste and mine water from the combine [21,36]. Figure 7 illustrates that due to the impact of the metallurgical plant there was an increase in concentrations of Ni from 191 to 129,916 mg/kg and Cu—from 34 to 22,965 mg/kg in sediments (0–13 cm) [21]. The increases in content of Co (up to 1498 mg/kg, while background level is 12 mg/kg), Zn (up to 376 mg/kg, background is 19 mg/kg), Cd (up to 18.8 mg/kg, background is 0.1 mg/kg), and Pb (up to 127.7 mg/kg, background is 1.4 mg/kg) were also noted. Moreover, there were some of the highest concentrations of chalcophile elements (Cd, Pb) of all lakes of Murmansk Region in this water body, which is related to the fact that pollutants enter Lake Nudjavr not only by air and through polluted soil, but also directly with wastewater from Kola MMC.

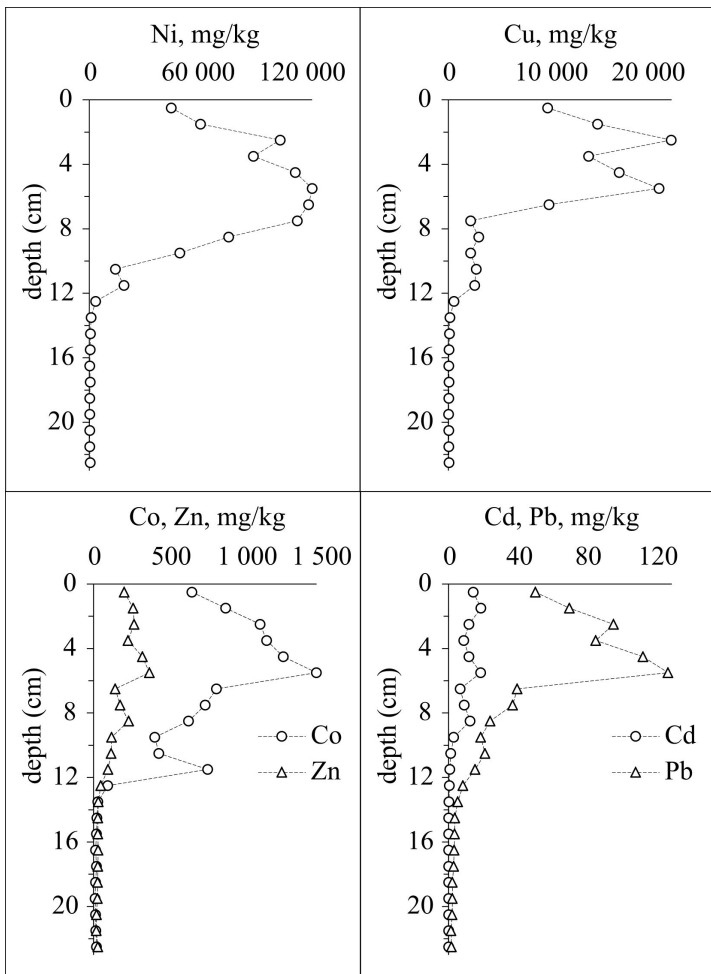

**Figure 7.** The vertical distribution of heavy metals in sediments of Lake Nudjavr (Murmansk Region) [21].

Studies of other lakes located in the impact zone of atmospheric emissions from the metallurgical plant [34] showed that there were similar dynamics of increased concentrations of heavy metals in all lakes, despite the different distances (from 7.5 to 12 km) from the source of pollution. Depending on the water body, the increases in the content of pollutants were found at depths of 10–15 cm, which indicated that the sedimentation rate in these lakes was ~2.3 mm/year. The concentrations of heavy metals in these lakes were by an order or even several orders of magnitude lower than in sediments of Lake Nudjavr and varied from ~500 to ~2200 mg/kg for Ni, from ~150 to ~1100 mg/kg for Cu, and from ~35 to ~115 mg/kg for Co [34]. The lowest concentrations of mentioned metals were found in Lake Pagel, located 12 km from the metallurgical plant.

Lake Imandra, on the bank of which the city of Monchegorsk is located, is also subject to the impact of Kola MMC. The most polluted area is Monche Bay, the part of the lake near the city and the metallurgical plant [35]. Here, the increases in concentrations of heavy metals (Ni, Cu, Co, Zn и Pb) were fixed at a depth of 10 cm, which corresponds to the start of operating of the combine in the late 1930s. The maximum contents of almost all mentioned pollutants were found in the uppermost layer of sediments (0–1 cm): ~16000 mg/kg for Ni, ~1400 mg/kg for Cu, ~315 mg/kg for Co, ~260 mg/kg for Zn, and ~75 mg/kg for Pb (Figure 8). Paleolimnological studies revealed that there are similar dynamics of behavior of main pollutants from the metallurgical plant in another part of Lake Imandra, Kunchast Bay, located ~100 km from Kola MMC [35], which may be related to both atmospheric and aquatic transport of substances in the largest water body of Murmansk Region. However, total values of heavy metals concentrations in sediments of Lake Imandra in

the area of Kunchast Bay were lower than in the area of Monche Bay. Thus, the maximum content of Ni in Kunchast Bay sediments was ~300 mg/kg, Cu—~120 mg/kg, Co—~25 mg/kg, Zn–~130 mg/kg, Pb–~36 mg/kg. Moreover, the increase in concentrations of pollutants began at a depth of 5 cm. Therefore, according to the knowledge of the timing of smelting/mining operations in the studied region, the sedimentation rate in these areas of Lake Imandra varied from ~0.8 to 1.6 mm/year.

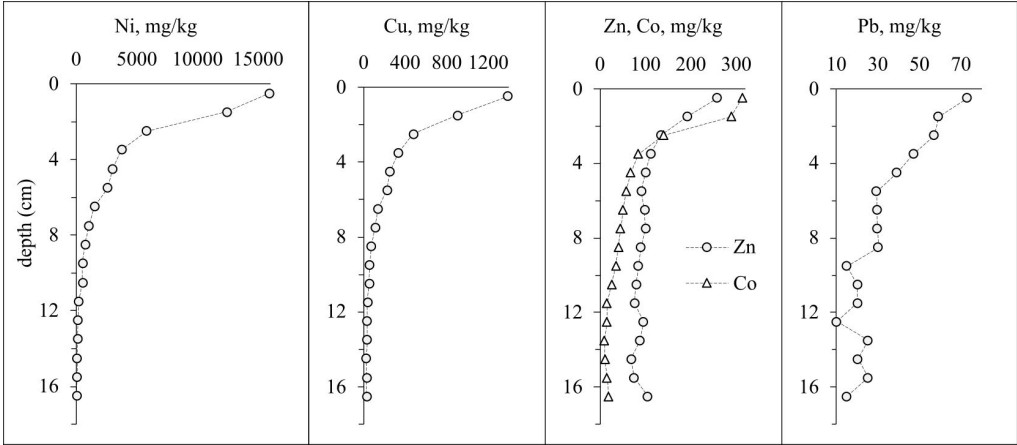

**Figure 8.** The vertical distribution of heavy metals in sediments of Lake Imandra (Murmansk Region) [35].

Another large industrial enterprise of Murmansk Region is Apatit JSC in Kirovsk, mining apatite-nepheline ore from the Khibiny deposit. Studies reported that wastewater and dust emissions from Apatit JSC played a significant role in the pollution of lakes located near this enterprise and its mines. The paleolimnological studies of Lake Bolshoi Vudjavr and Lake Imandra are the most illustrative [37–39]. Lake Bolshoi Vudjavr is the largest water body of the Khibiny Massif. This water body is mostly influenced by wastewater from mines. There were the increases in concentrations of P (up to ~15000 mg/kg, while background level is less ~1000 mg/kg), Ca (up to ~76500 mg/kg, background is ~1500 mg/kg), Sr (up to ~2900 mg/kg, background is ~400 mg/kg), Pb (up to 45.9 mg/kg, background is 9.7 mg/kg), and Cu (up to 225 mg/kg, background is 55 mg/kg) in upper layers of sediments of Lake Bolshoi Vudjavr [37,39–41], which is related to the composition of apatite-nepheline ore and also the influence of the city and the long-range transport of pollutants, including those from the metallurgical combine in Monchegorsk located ~45 km from this lake [42]. Recent studies of sediments of Lake Bolshoi Vudjavr have confirmed previously received data, broadened the range of identified elements, and specified the sedimentation rate in the lake [39,41]. Based on the data on the vertical distribution of the [210]Pb isotope, the sedimentation rate in the water body was 2.3 mm/year. Figure 9 illustrates that the increase in concentrations of heavy metals started in the early 1930s when the city of Kirovsk and the mining and concentrating company were founded. The majority of pollutants (Pb, Cu, Zn) enter the water body with wastewater from mines. However, V is probably related to the operations of boiler room, functioning until 2013, and has used heavy residual fuel oil (mazut). Additionally, increased contents of Sb and W in the sediments are related to the operations of the thermal power plant located ~10 km from the lake using coal as fuel [41].

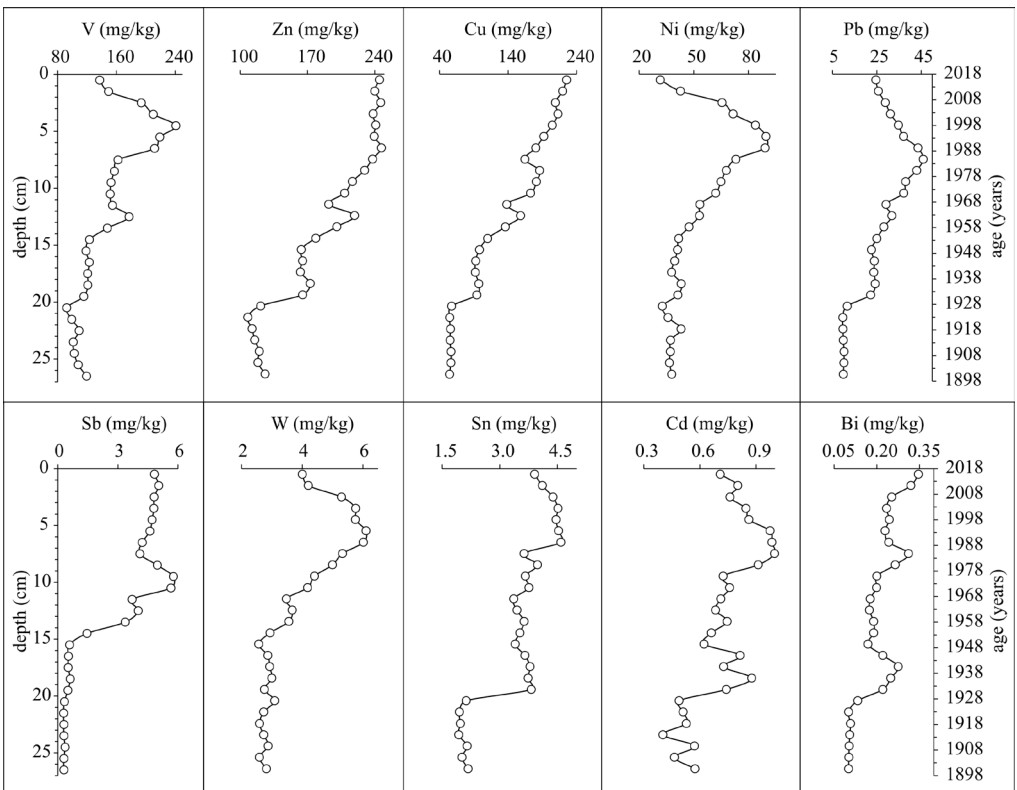

**Figure 9.** The vertical distribution of heavy metals in sediments of Lake Bolshoi Vudjavr (Murmansk Region) [39,41].

Besides the anthropogenic impact on Lake Bolshoi Vudjavr, paleolimnological studies determined the natural geochemical anomaly of Mo in sediments of the studied lake [39,43]. It was shown that the sediment cores were enriched with Mo both in upper layers (up to 9.9 mg/kg) due to the influence of mine waters and lower layers (up to 15.1 mg/kg) due to the influence of underlying rocks with increased concentrations of this metal. Previously, the increased concentrations of Mo were found in rivers, streams, and industrial wastewater entering Lake Bolshoi Vudjavr [44].

The impact of the Apatit JSC operations was also revealed in Lake Imandra, the largest lake of Murmansk Region. This was reflected in the upper layers of sediments in the increased content of P, Ca, Sr, and rare earth elements [38], enriching rocks in the Khibiny Massif [45]. Paleolimnological studies determined the increase in concentrations of rare earth elements in sediments of Lake Imandra at a depth of 10 cm, which corresponds to the start of operating of the ore-processing plant. The highest concentrations of rare earth elements (up to ~240 mg/kg for La (background is 56.5 mg/kg) and up to ~400 mg/kg for Ce (background is 80.6 mg/kg)) in studied sediments date back to the 1970s, the period of the most active ore production (Figure 10) [38]. Even higher concentrations of rare-earth elements due to the activities of JSC Apatit were found in the upper layers of the sediments of Lake Bolshoy Vudjavr [39]. For instance, the detailed analysis of the sediment core of this lake revealed a tendency towards increased concentrations of La (up to 535 mg/kg, while minimum in the core is 84 mg/kg), Ce (up to 802 mg/kg, while minimum in the core is 128 mg/kg), Sm (up to 44 mg/kg, while minimum in the core is 7.3 mg/kg), and Eu (up to 13 mg/kg, while minimum in the core is 2). Obviously, due to the close proximity of Lake Bolshoy Vudjavr to the plant, the concentration of rare earth elements in the sediments of this lake is significantly higher than in Lake Imandra.

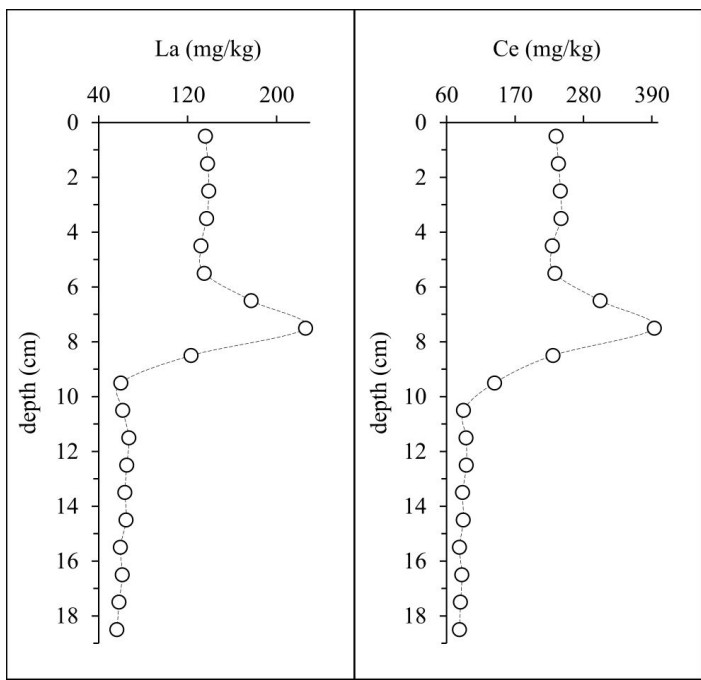

**Figure 10.** The vertical distribution of La and Ce in sediments of Lake Imandra (Murmansk Region) [38].

The studies of lakes of Murmansk Region, including Lake Imandra, located in the impact zone of Olcon JSC, mining, and processing iron-bearing ores, demonstrated the increase of the contents of Fe (up to 20%, while background level is ~3%) and Mn (up to 4%, background is ~1%) in the upper layers of studied sediments [46]. Moreover, the pollution of these lakes by Kola MMC was fixed using marker elements Ni, Cu, and Co with their increased concentrations in 0–10 cm layers similar to other studies. The average sediment rate in studied lakes was 1–2 mm/year, which is close to the average sediment rates in lakes of Murmansk Region in the industrial period [47].

### 3.1.3. Conclusions of Section 3.1

Anthropogenic influence is considered on the example of the Murmansk region and the Ural region (mainly the Chelyabinsk region). The impact on lake ecosystems from metallurgical enterprises and the mining industry is shown. Lake sediments formed during the 20th and 21st centuries are characterized by a significant level of enrichment in heavy metals (Ni, Cu, Zn, Co, Mo, Pb, Cd) and other elements (for instance, rare earth metals).

### *3.2. Urbanized Areas*

A great number of potential sources of anthropogenic pollution are often concentrated in cities. These sources are industrial enterprises, all means of transport, road, and construction dust, and household waste [48–52]. Moreover, the long-range transport of pollutants influences the city areas similar to other (non-urban) areas. The targeted detailed paleolimnological studies of urban areas in Russia were conducted only by the author and his colleagues from the Institute of the North Industrial Ecology Problems of Kola Science Center of RAS and the Institute of Geology, Karelian Research Centre of RAS in Murmansk Region and the Republic of Karelia. It should be noted that these studies are still ongoing.

According to different monitoring services, the Republic of Karelia is one of the clean regions of Russia. There, the anthropogenic pollution of the aquatic environment is mainly related to urban areas and rarely to industrial areas [53]. The majority of paleolimnological studies of the anthropogenic impact on lakes were conducted in Petrozavodsk, the largest

city of Karelia [54,55]. For instance, the detailed analysis of the sediment core of Lake Lamba located in the northern part of the city district revealed a tendency towards increased concentrations of heavy metals, including Pb (up to 137 mg/kg, while background is 4 mg/kg), Cd (up to 1.2 mg/kg, background is 0.2 mg/kg), Ni (up to 607 mg/kg, background is 22 mg/kg), V (up to 4785 mg/kg, background is 17 mg/kg), Cr (up to 179 mg/kg, background is 10 mg/kg), Cu (up to 1189 mg/kg, background is 45 mg/kg), Zn (up to 963 mg/kg, background is 136 mg/kg), etc. (Figure 11). The analysis of concentrations of mentioned elements in lower (Holocene) layers of sediments showed that they were similar to the background, or even lower [56].

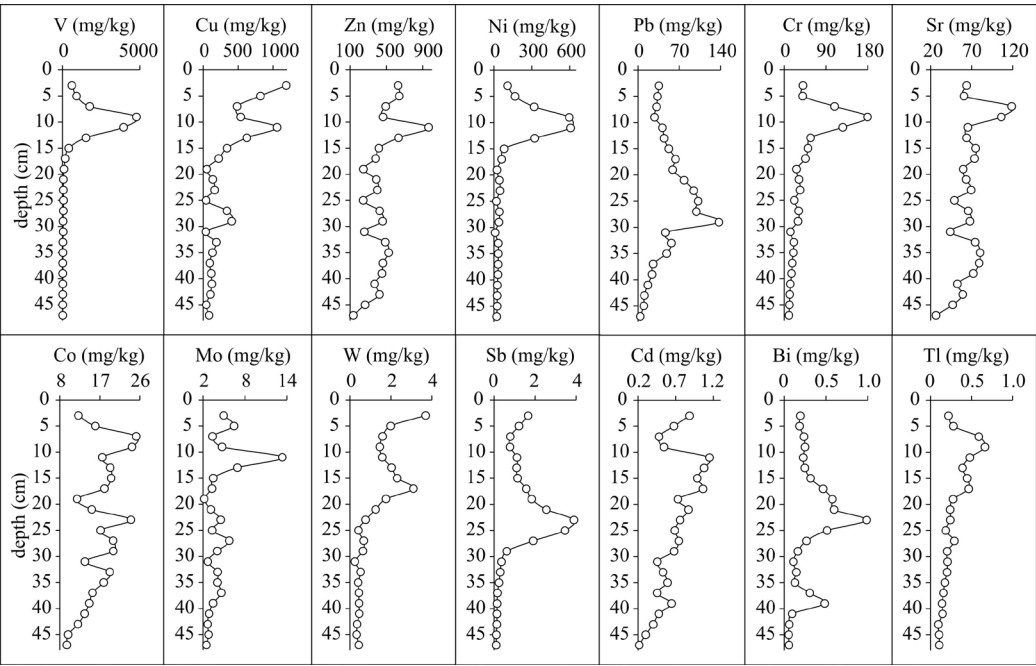

**Figure 11.** The vertical distribution of heavy metals in sediments of Lake Lamba (the Republic of Karelia) [55].

The analysis revealed that the lake was exposed to the multifactorial outside load. For example, the increased levels of V and Ni are related to the operations of the thermal power plant (Figure 12), which had been using mazut from 1978 until 2000 and then started using gas. Studies show that mazut boiler rooms and thermal power plants always induce increased concentrations of V and Ni in the environment [57,58]. The transition of the Petrozavodsk thermal power plant from mazut to gas resulted in a sharp decrease in concentrations of both heavy metals in the uppermost layers of sediments of Lake Lamba (Figure 11). The increase in concentrations of Zn, Cu, W, and Mo in sediments of the water body is associated with the operations of engineering and instrument-making plants [59], and Pb is related to transport, which had used leaded gasoline with tetraethyllead in Russia until 2002 [13,18,60,61]. Similar behavior of mentioned heavy metals was observed in the paleolimnological study of Lake Chetyrekhverstnoe, also located in the Petrozavodsk city area [54]. The exception was V and Ni behavior. Concentrations of these metals were significantly lower in sediments of Lake Chetyrekhverstnoe compared to Lake Lamba, as the first lake (the Chetyrekhverstnoe) is located 11 km from the thermal power plant and the other is 500 m from the plant. It is known that the range of transfer of particles from mazut thermal power plants and boiler rooms usually does not exceed ~15 km [62].

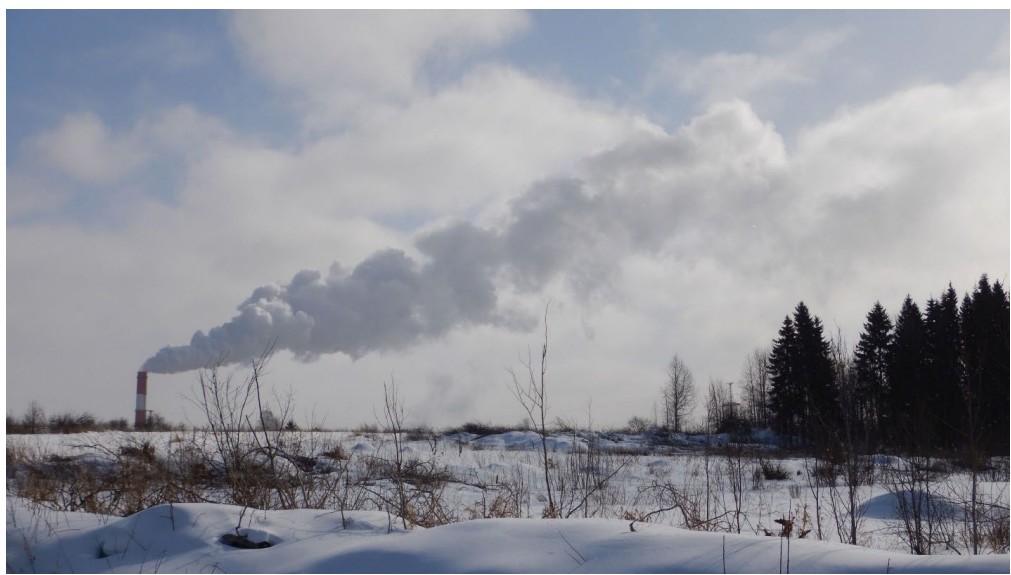

**Figure 12.** The view of the Petrozavodsk thermal power plant (the Republic of Karelia).

Based on the paleolimnological studies, in other cities of the Republic of Karelia (Medvezhyegorsk, Suoyarvi, Sortavala) the main pollutants are Pb, related to the transport activities and the long-range transport of pollutants [63], Sb and Cd, entering lakes due to fuel combustion all around the world [64], and rarely Cu, Zn, and Sn, which may be related to both transport and dust pollution of urban areas [65–67]. The main geochemical markers allowed for determining that the sedimentation rate in urban lakes of Karelia varied from 2 to 5 mm/year [66].

The impact of urbanized areas in Karelia was also shown in the analysis of geochemistry of sediments of Vygozero Reservoir located in the center of this region [68,69]. The increase in concentrations of V (up to 171 mg/kg, while background for Vygozero is 48 mg/kg), Ni (up to 46 mg/kg, background for Vygozero is 26 mg/kg), Pb (up to 26 mg/kg, background for Vygozero is 6.4 mg/kg), Cd (1.3 mg/kg, background for Vygozero is 0.7 mg/kg), and Bi (up to 0.28 mg/kg, background for Vygozero is 0.09 mg/kg) at depths of 16–24 cm was fixed in the sediment core sampled ~5 km from the city of Segezha (Figure 13). Considering that recent sediments of Vygozero Reservoir have formed in the last 80 years, the age of the studied core was no more than 30–40 years. Therefore, the newest anthropogenic processes relevant to the environment of Segezha were fixed in this case. For instance, the increased level of V and Ni accumulation is associated with emissions from the Mazut thermal power plant operating since the 2000s [69]. The increase in concentrations of Pb, Cd, and Bi in the uppermost layers of sediments of Vygozero Reservoir is evidence of the perpetual entering of pollutants into the area of the North of Russia due to the long-range atmospheric transport [70,71].

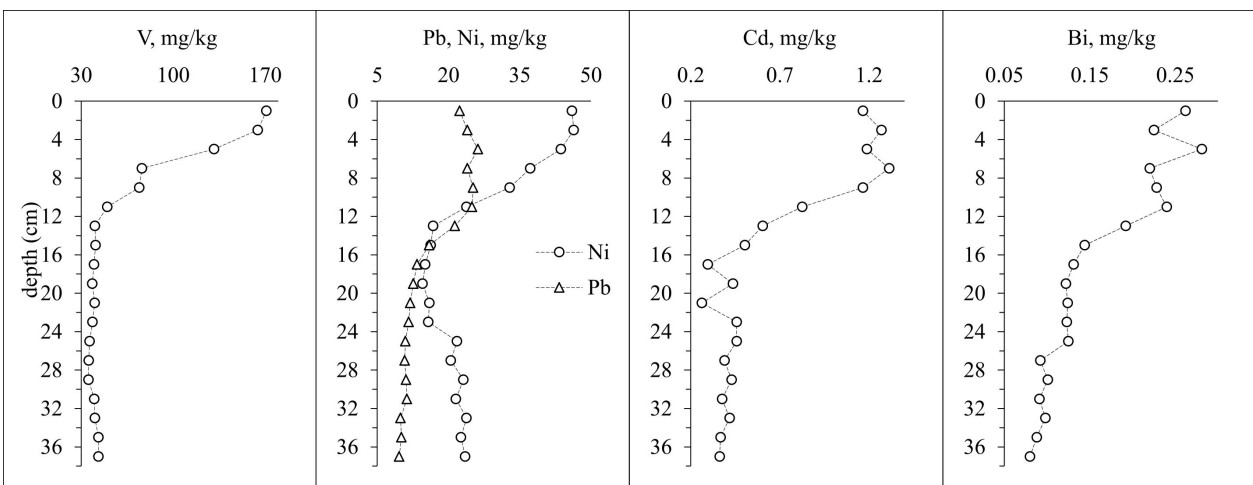

**Figure 13.** The vertical distribution of heavy metals in sediments of Vygozero Reservoir (Republic of Karelia) [69].

Murmansk Region is one of the most urbanized regions of Russia. However, paleo-limnological studies in the city areas of this industrial region have not been conducted until recently, since they were focused mainly on the impact of the metallurgical plants and mining enterprises. Similar to Karelia, the urban lakes of Murmansk Region are subject to pollution from the energy industry, transport, and also the metallurgical plants mentioned before [52,72,73].

In the city of Murmansk, which is the capital of Murmansk Region, the analysis of the recent sediment core of Lake Semenovskoe (Figure 14) showed the increases in concentrations of heavy metals starting from a depth of 32 cm for Pb (up to 125 mg/kg, while background is ~4 mg/kg), 28 cm for Zn (up to 694 mg/kg, background is ~76 mg/kg), Co (up to 38 mg/kg, background is ~5 mg/kg), Ni (up to 263 mg/kg, background is ~27 mg/kg), Cd (up to 3.2 mg/kg, background is ~0.3 mg/kg), and Sb (up to 4.1 mg/kg, background is ~0.08 mg/kg), and 16 cm for V (up to 904 mg/kg, background is ~70 mg/kg). Studies [74,75] revealed that the Mazut thermal power plant and boiler rooms play a significant part in the pollution of Murmansk lakes with V and Ni, because mazut has been used at this enterprise since the 1960s as the main fuel [76]. Based on the dynamics of behavior of the two mentioned pollutants, it was determined that the average sedimentation rate in the lake in the industrial period was ~3 mm/year. Other metals are related to dust emissions from the coal terminal in the Murmansk port (Zn, Co, Pb, Cu, Cd, Sb), transport using leaded fuel (Pb) [5], the incineration plant, and also the influence of the long-range transport of pollution from the local plants and the plants located in other regions of Russia and other countries [64,77]. It should be noted that all studied lakes of Murmansk are characterized by similar dynamics of behavior of mentioned heavy metals [72].

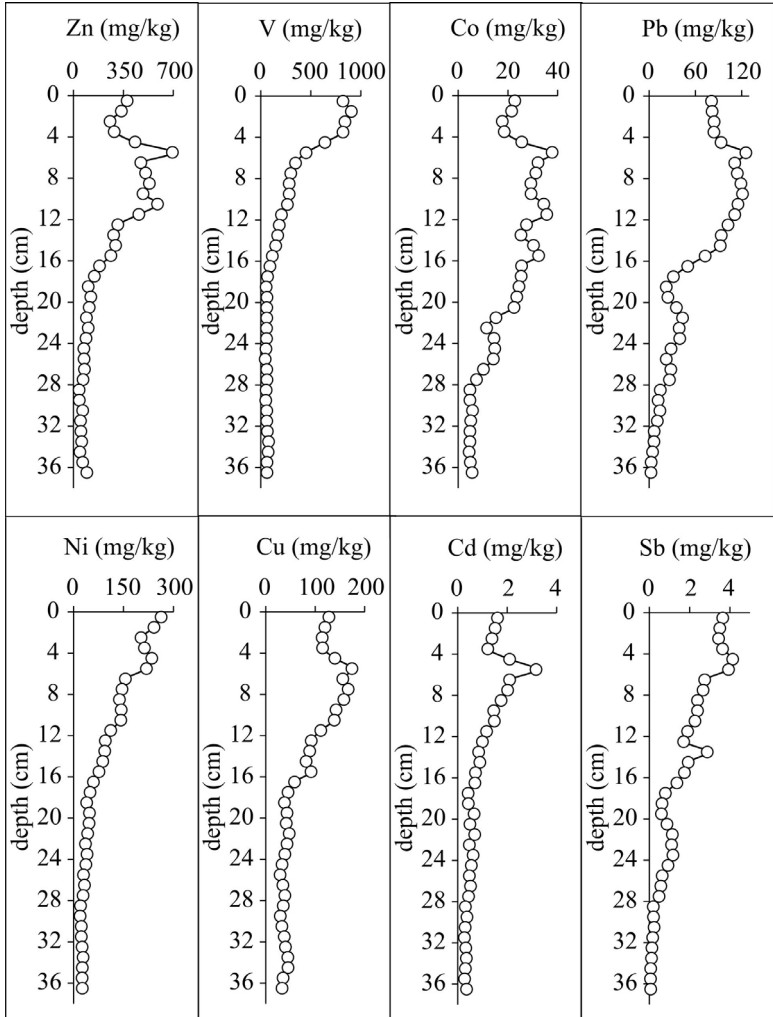

**Figure 14.** The vertical distribution of heavy metals in sediments of Lake Semenovskoe (Murmansk Region) [52].

In addition, studies of the urban lakes of Murmansk showed that rare earth elements can also be indicators of technogenic impact on water bodies [78]. In the course of the work, the general dynamics of the accumulation of rare earth elements and «classical» heavy metals in the upper layers of sediments of polluted lakes were established. Basically, rare earth elements enter aquatic ecosystems as a result of dust emissions (from transport, enterprises, wear of buildings and roads, destruction of soil cover and rocks) [49,79,80]. Similar patterns have not been established in the remote territories of the Murmansk region, since the described processes have a minimal manifestation there.

In Monchegorsk, the other city of Murmansk Region mentioned before, the main anthropogenic load on Lake Komsomolskoe comes from Kola MMC emissions [81]. The stable dynamics of increased concentrations of a wide range of heavy metals such as Ni (up to 2140 mg/kg, while the background is 89 mg/kg), Cu (up to 2607 mg/kg, background is 68 mg/kg), Cr (up to 335 mg/kg, background is 54 mg/kg), Zn (up to 335 mg/kg, background is 41 mg/kg), Co (up to 129 mg/kg, background is 4 mg/kg), V (up to 140 mg/kg, background is 35 mg/kg), Pb (up to 100 mg/kg, background is 8 mg/kg), Cd (up to 2.5 mg/kg, background is 0.4 mg/kg), Sb (up to 3.3 mg/kg, background is 0.2 mg/kg), etc., were fixed in recent sediments of this lake [82]. Similar tendencies of heavy metals behavior, shown earlier on the example of other lakes in the impact area of the plant, remain there, mainly because Lake Komsomolskoe is located 4 km from the metallurgical plant [34,35]. Besides, the impact of the Mazut thermal power plant located on the premises of the metallurgical plant was noted for the first time using marker element V. The average

sedimentation rate, calculated using $^{210}$Pb isotope activity in this urban lake, was 2.7 mm/year [82]. The comparison of the age of sediments and the dynamics of behavior of heavy metals showed that the increase in main pollutants content began in the late 1930s when the plant near Monchegorsk started operating.

Other urbanized areas of Russia are poorly studied from the paleolimnological point of view. Unfortunately, despite the great activity of lake researchers in the Republic of Tatarstan and a large number of publications on the content of heavy metals in recent lake sediments [83–85], there are almost no studies with the detailed analysis (layers from 0 to 10 cm) of the vertical distribution of pollutants in sediment cores. There is only one example of such a study of the urban water body in the Republic of Tatarstan. In particular, the studies of the geochemistry of the sediment core 110 cm long of Lake Verkhny Kaban located in the city of Kazan revealed the anthropogenic impact on the lake by the vertical distribution of Pb (up to 45 mg/kg, minimum for the sediment core is 6.1 mg/kg), Cd (up to 4.7 mg/kg, minimum for the sediment core is 0.01 mg/kg), Cu (up to 176 mg/kg, minimum for the sediment core is 0.2 mg/kg), and Zn (up to 480 mg/kg, minimum for the sediment core is 1.4 mg/kg) [86]. The highest concentrations of mentioned heavy metals were found in the upper layers of sediments accumulated in the area of the discharge channel of the thermal power plant. Moreover, other industrial enterprises of Kazan use this channel for untreated water disposal.

Conclusions of Section 3.2

According to published data, paleolimnological studies of the anthropogenic impact of cities on the environment were carried out on the example of urbanized areas of the Republic of Karelia and the Murmansk region (north-west Russia). The pollutants of water ecosystems are industrial enterprises, thermal power plants, boiler houses, waste processing plants, and transport (primarily cars). In recent sediments of lakes, background excesses for V, Ni, Pb, Zn, Cu, Cd, and others have been established. In addition, it was found that lithophile elements that enter the environment with dust from the destruction of soil, road surfaces, and buildings can also be indicators of urban impact on water bodies.

*3.3. Background (Pristine) Areas*

The important part of paleolimnological studies in Russia is the study of lakes in the regions remote from the anthropogenic sources of pollution. To a certain extent, these regions can be considered as a background. First of all, this concerns the Arctic zone of the Russian Federation. The studies of the levels of heavy metals accumulation in sediments of lakes in such areas are of interest, mainly in terms of the study of the long-range transport of pollutants [3,9,18,70,87].

Udachin V.N. and his colleagues conducted studies of the arctic lake Kenteturku located in the center of the Taimyr Peninsula [88]. Researchers sampled the sediment core 30 cm long and divided it into 1 cm layers. It was found that the lake is still practically pristine. There was no significant exceedance of heavy metals concentrations in the upper layers of studied sediments, except for Pb in the 1–2 cm layer (Figure 15).

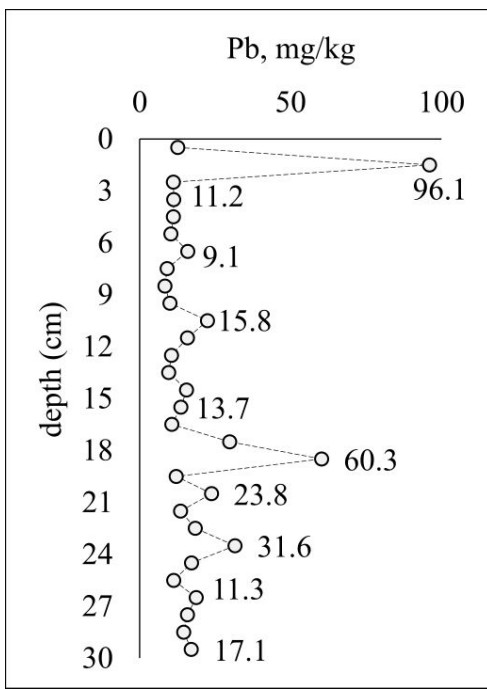

**Figure 15.** The vertical distribution of Pb in sediments of Lake Kenteturku (the Arctic) [88].

First of all, it is interesting to note that there were no abnormal peaks of Ni and Cu concentrations, considering that the Norilsk industrial hub is located 550 km from the lake [89]. Emissions from the metallurgical plant are likely to not reach the studied area and Lake Kenteturku. The peak concentration of Pb (96 mg/kg), in contrast to the median content (14 mg/kg) throughout the sediment core, seems to be a measurement error on the one hand. However, on the other hand, taking into account possible extremely low sedimentation rates in the lake, the sharp increase in Pb concentrations may indicate the influence of the long-range transport of pollutants, which is typical for the recent lake sediments in the Northern Hemisphere [14,34].

The studies of the geochemistry of sediments of other arctic lakes located in the Yamal and the Gyda Peninsulas also did not show the significant dynamics of the majority of elements except for Hg [90]. In sediments of Lake Langtibeito, the concentration of this metal slightly increased to ~0.08 mg/kg, starting from a depth of ~10 cm. The sedimentation rate in lakes of mentioned areas was from 1.7 to 2.0 mm/year based on the $^{210}$Pb activity.

In the Murmansk Region, which also belongs to the Arctic zone, the studies of lakes of the background areas showed a tendency towards an increase in traditional pollutants from among chalcophile elements (Pb, Cd, Hg, As) and local pollutants Ni and Cu from Kola MMC emissions in the uppermost layers of sediments [34,91]. The increased concentrations of Ni (from 12 up to 111 mg/kg) and Pb (from 8 up to 36 mg/kg) in the uppermost layers (5–6 cm) of sediments were even found in the lakes located in the mountainous areas, which can act as a barrier for pollutants distribution [92]. The sedimentation rate in such lakes can be assessed as ~1 mm/year or less based on the marker pollutants. In Lake Umbozero, the second largest lake of Murmansk Region, concentrations of heavy metals in sediments increased from a depth of ~10 cm (typical for Pb, Cd, As) and ~5 cm (typical for Ni and Cu) [14,93]. In total, both the largest lakes of Murmansk Region are characterized by similar patterns of accumulation of heavy metals, which are the main pollutants in the region.

In recent years, when studying the lakes of Murmansk Region, a range of pollutants also indicating the long-range atmospheric transport has been extended due to the use of new methods for the analysis of microelements in sediments (ICP-MS) [94]. In lake sediments of pristine areas of the northern part of Murmansk Region (the area of the Rybachy

Peninsula), the increased concentrations of Ni (up to 127 mg/kg, background is 25 mg/kg), Cu (up to 370 mg/kg, background is 35 mg/kg), Co (up to 40 mg/kg, background is 4.2 mg/kg), Pb (up to 82 mg/kg, background is 10 mg/kg), As (up to 6.3 mg/kg, background is 1.8 mg/kg), Sn (up to 32 mg/kg, background is 0.8 mg/kg), Bi (up to 1 mg/kg, background is 0.06 mg/kg), Sb (up to 0.5 mg/kg, background is 0.08 mg/kg), and Tl (up to 0.11 mg/kg, background is 0.09 mg/kg) were found. Paleolimnological studies of lakes in the south of Murmansk Region and the north of the Republic of Karelia showed that the range of transport of emissions from Kola MMC enterprises reached ~250 km [82]. The sedimentation rate in lakes of pristine taiga landscapes of Northwest Russia can be ~0.6 mm/year based on the $^{210}$Pb isotope [82].

Similar tendencies of the behavior of the above-mentioned heavy metals in lake sediments are found in two regions located to the south of Murmansk Region. These are the Republic of Karelia [54,95] and Arkhangelsk Region [96]. In the study of the geochemical analysis of recent sediments of Lake Maselgskoe (the south-west of Arkhangelsk Region), it was determined that the upper layers of sediments were enriched with Pb, Cd, Sb, Bi, and W. Particularly, the increase in concentrations of Pb (up to ~50 mg/kg) started at a depth of ~30 cm. The sedimentation rate was 4.1 mm/year based on the nonequilibrium $^{210}$Pb [96]. Other studied elements (for instance, Sc and Zn) do not tend to increase in the upper layers of sediments of Lake Maselgskoe, since they are not the agents of the long-range atmospheric transport.

The sedimentation rate in recent sediments of Lake Ukonlampi located in the south-eastern part of Karelia (near the Finnish–Russian border) was 1.25 mm/year based on the $^{210}$Pb activity [97,98]. The similar tendency towards increased concentrations of Pb (up to 91.1 mg/kg, background is 3.8 mg/kg), Cd (up to 2.69 mg/kg, background is 0.39 mg/kg), Sb (up to 1.97 mg/kg, background is 0.10 mg/kg), Sn (up to 5.34 mg/kg, background is 0.46 mg/kg), Tl (up to 0.84 mg/kg, background is 0.06 mg/kg), Bi (up to 4.06 mg/kg, background is 0.08 mg/kg), Cu (up to 51.2 mg/kg, background is 12 mg/kg), and Zn (up to 263.8 mg/kg, background is 40 mg/kg) was found in this lake and two water bodies nearby (Figure 16). It should be noted that pollution of these background water bodies might be associated not only with the global pollution of the Northern Hemisphere but also with the proximity of this region to industrial enterprises of Finland in Imatra and Kotka [99]. This explains the increased content of Zn and Cu, which usually are not categorized as indicators of the atmospheric transport in pollution of the North background regions. In total, the analysis of recent sediment cores of 30 small lakes of the south of Karelia and Vygozero Reservoir showed that the main pollutants in the region are due to the long-range atmospheric transport of Pb, Sb, Cd, Bi, and Tl [67,69]. The close correlation between concentrations of these metals (for instance, Pb and Sb, Figure 17) confirmed the unity of their entering to the aquatic ecosystem and accumulation in lake sediments.

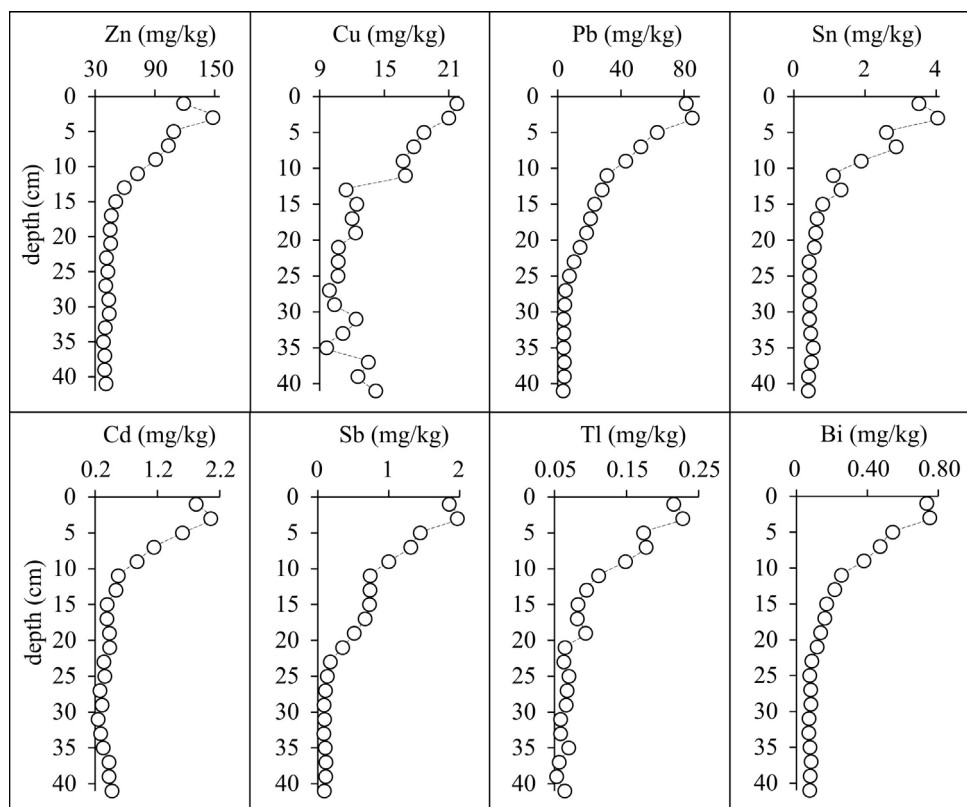

**Figure 16.** The vertical distribution of Pb in sediments of Lake Liunkunlampi (the Republic of Karelia) [98].

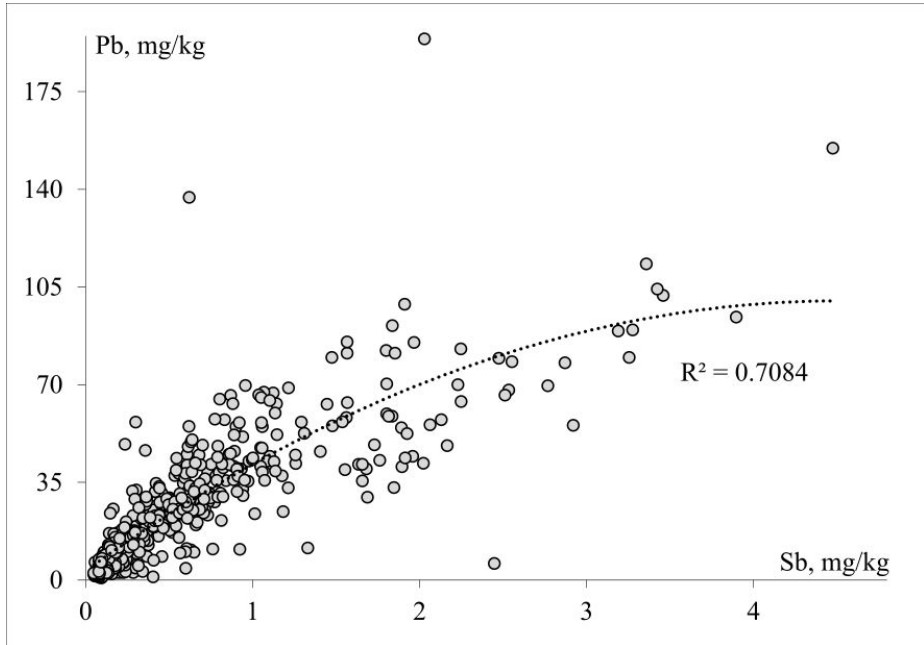

**Figure 17.** The correlation between Pb and Sb in lake sediments of Karelia (author data).

Similar patterns are observed in Lake Onega, the largest reservoir in the Republic of Karelia and the second largest in Europe [95]. In technogenesis, the rate of sedimentation in Lake Onega is not high, 1 mm/year, which is estimated from the activity of [137]Cs and [210]Pb isotopes [100]. This value is close to the average sedimentation rates in lakes in remote (background) areas. It has been established that the content of Pb, Cd and Sb increases in the upper (up to 10 cm) sediment layers of Lake Onega. The authors attribute

this dynamic to the technogenic impact on the lake, primarily due to the long-range transport of pollutants. In particular, an increase in the concentration of Pb up to ~40 mg/kg (with a background level is ~10) and Cd up to ~1 mg/kg (background is ~0.2) was found.

Siberia is a region of Russia where limnological geochemical studies of freshwater sediments are well-developed. The focus of Siberian paleolimnology has been on the analysis of natural variations of chemical elements in sediment cores. At the same time, there are studies on the determination of the anthropogenic impact on lake ecosystems and the environment.

The study of the sediment core of Lake Manzherok in the Altai Republic (Siberia) showed the difference in the accumulation of heavy metals such as Pb (up to ~13 mg/kg, minimum for the core is ~4.5), Cd (up to ~1.7 mg/kg, minimum for the core is ~0.15), and As (up to ~35 mg/kg, minimum for the core is ~4) in the uppermost layers of sediments (0–20 cm) compared to lithophile elements [15]. Paleolimnologists suggest that this behavior of the mentioned heavy metals can be explained by the anthropogenic impact on the studied lake and its catchment area. In another Siberian lake (the Kolyvanovskoe) located in the southwest of Altai Krai, similar behavior of Pb, Cd, and Hg was observed in recent sediments [101]. The concentrations of mentioned heavy metals increased from the lower to the upper layers in the 50-cm core dated using [137]Cs and [210]Pb isotopes. The age of studied sediments of Lake Kolyvanovskoe showed that the increase in the concentrations of Pb (up to ~25 mg/kg, background is ~9), Cd (up to ~0.3 mg/kg, background is ~0.05), and Hg (up to ~0.3 mg/kg, background is ~0.02) started in the period from the end of the 19 century to the present. Other trace elements such as Cu, Co, Zn, and Ni do not have similar accumulation dynamics in the studied sediments of Lake Kolyvanovskoe.

The extensive studies of Siberian lakes demonstrate common patterns of increased concentrations of Pb, Cd, Hg, and Sb in sediments dated back to the last three centuries using [137]Cs and [210]Pb isotopes [17,102,103]. The majority of scientists admit the significant anthropogenic impact on the formation of geochemical anomalies of mentioned elements [101,104]. The high concentrations of some heavy metals were found in recent sediments of small lakes: 3345 mg/kg of Pb in sediments of Lake Bolshye Rakity, adjacent to the city of Rubtsovsk, 112 mg/kg of Sb, and 4.2 mg/kg of Cd in sediments of Lake Yakov (Tomsk Region) [17]. However, such concentrations of heavy metals are not common for small lakes of Siberia, even in cases of the anthropogenic impact on studied lakes. For instance, in the uppermost layers of sediments of Lake Kotokel (Pribaykalsky District), the concentrations of Pb reached 11.5 mg/kg (minimum for the core is 6.4) and Cd—0.4 mg/kg (minimum for the core is 0.13) (Figure 18) [104]. In the uppermost layers of sediments of Lake Shchuchie (Tomsk region), Cd concentrations reached 0.83 mg/kg (minimum for the core is 0.08). Despite the historical dynamics of the anthropogenic input of heavy metals into the water bodies, median background levels of Pb (20 mg/kg), and Cd (0.14 mg/kg) for Siberia, in the lake, sediments are not often exceeded [17].

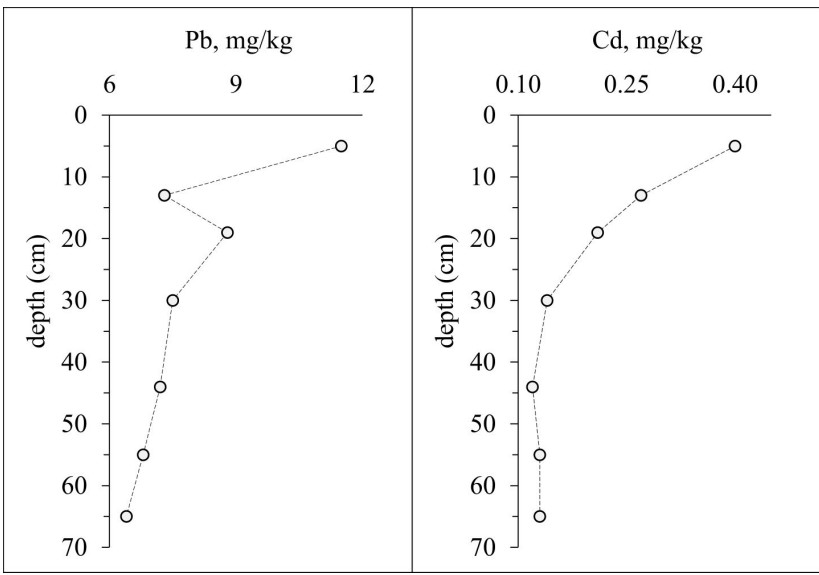

**Figure 18.** The vertical distribution of Pb and Cd in sediments of Lake Kotokel (Siberia) [104].

Conclusions of Section 3.3

The geography of paleolimnological studies of the anthropogenic impact on the pristine areas of Russia is more extensive than in the previous sections of the article. Studies of lake sediments were carried out here in the Arctic, in the taiga zone of Karelia, and in different regions of Siberia. Practically everywhere, the influence of the long-range transport of heavy metals (Pb, Sb, Cd, Bi, Tl) associated with the combustion of fossil fuels at the enterprises of North America, Europe, and Asia is manifested. According to isotopic dating, low sedimentation rates are noted in the lakes of the background areas compared to industrial and urban areas.

### 3.4. Comparison of Element Concentrations

In order to compare the values of the content of chemical elements in the sediments of lakes from different regions of Russia, it was necessary to carry out a normalization procedure. For this, the average content of chemical elements in the upper part of the Earth's crust [105] was used, by which the concentrations of elements in the sediments of the lakes were divided (the uppermost layers of the lake cores were taken). After that, the obtained data (enrichment factors) were logarithmic so that they could be placed on one chart (Figure 19).

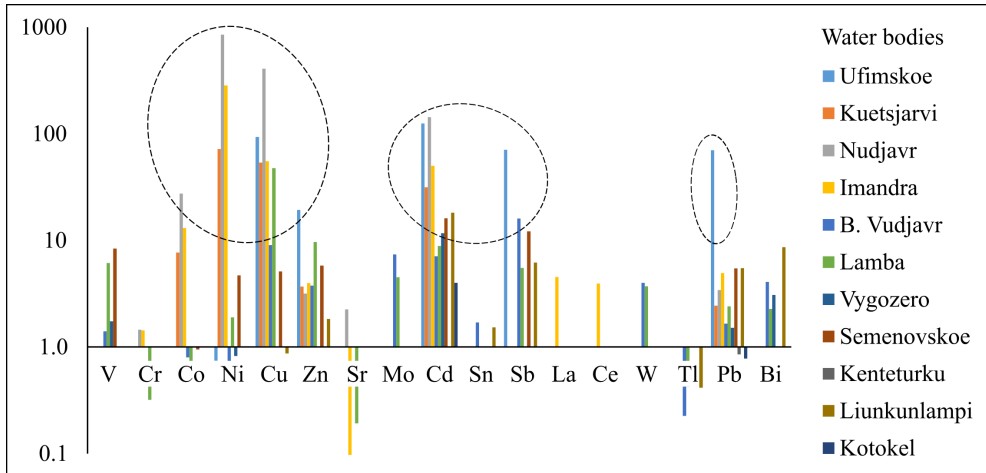

**Figure 19.** Enrichment factors of chemical elements from lake sediments described in this review (Figure 1).

It can be seen that the sediments of industrial lakes (Ufimskoe, Kuetsjavri, Nudjavr) are the most polluted. The highest enrichment factors are noted for Ni, Cu, Cd, Pb, and Sb. On the other hand, even in the lakes of the background areas, there are excesses of heavy metals, which are associated with the long-range atmospheric transport of pollutants (Cd, Pb, Bi, Sb). Vanadium is a specific pollutant in the urban lakes of the North-West of Russia, which, as noted, is associated with emissions from mazut thermal power plants and boiler houses.

## 4. General Conclusions and Perspectives

### 4.1. Conclusions

The analysis of the large number of paleolimnoligical studies of the recent anthropogenic impact on lakes of Russia showed that, despite the large distances between regions, likely different geology, and other factors influencing the sedimentation, there were a lot of similarities in the accumulation dynamics of pollutants in lake sediments. For instance, the specifics of metallurgical and mining plants are well fixed both in Chelyabinsk Region (the Southern Ural) and Murmansk Region (Northwest Russia). The increases in Cu, Zn, and Pb concentrations in the upper layers of sediments were observed in lakes near the metallurgical plant of the city of Karabash [16]. Moreover, this enterprise influences even lakes located at a distance of 100 km from emissions. A similar situation can be observed in Murmansk Region [21,27], where Cu and Ni are the key pollutants. Sediments most polluted with these metals were found in Lake Kuetsjarvi (the area of the Norway–Russia border) and Lake Nudjavr (the central part of the region). In Murmansk Region, paleolimnological studies of the impact of the mining enterprises determined the increased concentrations of P, Ca, Sr, and rare earth elements in recent sediments, as all these elements are included in produced ore entering water bodies with mine waters and dust [39].

The similarity in the impact of Mazut boiler rooms and thermal power plants on sediments of lakes of Petrozavodsk, Segezha, Murmansk, and Monchegorsk was found in urban areas of Karelia and Murmansk Region. It was shown that all water bodies were enriched with V and Ni, included in ash from mazut burning [54]. Moreover, there were increased concentrations of Pb in sediments of these cities due to the active use of leaded fuel in cars all over the world [61]. In Russia, the fuel containing Pb was banned in 2002. Besides, the impact of engineering and instrument-making companies was fixed in sediments of Petrozavodsk lakes, and the impact of the coal port, the incineration plant, and metallurgical industry was observed in lakes of Murmansk Region [72,74].

The special analysis of areas not subject to the direct anthropogenic impact showed that lake sediments in the Arctic zone of Russia, Karelia, Arkhangelsk Region, and Siberia were still influenced by the long-range transport of pollutants. Mostly, it is related to the burning of fossil fuel (coal), therefore Pb, Cd, Sb, Hg, Bi, and Tl, included in coal as additives, are the main geochemical agents of this process [3,87]. The clearest patterns of the increase in concentrations of these heavy metals are found in Northwest Russia, possibly due to its proximity to Europe [97]. The studies of a great number of lakes of the Republic of Karelia showed that Pb is closely associated with Sb in sediments of this region, which indicates the similar pattern of the input and accumulation of metals in sediments of lakes in pristine areas [67].

Studies also demonstrated that sedimentation rates estimated using $^{210}$Pb and $^{137}$Cs isotopes varied to a large extent in modern times. The lowest sedimentation rates (less than 1 mm/year) were fixed in small lakes of background areas or in large water bodies of the Russian North [92]. The highest sedimentation rates (from 3 to 5 mm/year) were found in lakes of urban or industrial areas [22]. In regions subject to the increased level of the anthropogenic load, the more intensive weathering, together with the atmospheric inputs of pollutants to water, possibly lead to the larger amount of matter accumulated in lakes.

## 4.2. Perspectives

The review of all known paleolimnological research aimed at the influence of the modern anthropogenic load on the environment of Russia showed that all these studies were concentrated in three regions—the Northwest, the Urals, and Siberia. In the author's opinion, it is related not only to the fact that there are a lot of lakes and several large anthropogenic objects in these regions, but also to the lack of the necessary equipment for the detailed sediment core sampling and the analysis of a wide range of chemical elements including heavy metals, and the lack of human resources for conducting paleolimnological studies in other regions. For instance, despite the fact that a lot of studies of geochemistry of lake sediments are conducted in order to analyze the anthropogenic impact on water ecosystems in the Republic of Tatarstan [86], there are almost no works with the detailed analysis of sediment cores of urban and remote lakes. Unfortunately, there are no such works in other regions of Russia, where they can be highly in demand. These studies can be relevant for Moscow, Saint Petersburg, large cities of Siberia, and the Russian Far East. Recently, the author conducted detailed studies of the sediment cores of lakes of Arkhangelsk, which will be published soon. However, this is obviously not enough, considering that there are a lot of significant regions of Russia where paleolimnoligical studies of the anthropogenic load on the environment have not yet been conducted. Hopefully, such works will be done in future involving international cooperation, since the equipment for sampling recent sediments with an option of the detailed dividing cores into layers is mainly produced abroad (for instance, in Finland and Norway), and lake research is almost always included in European scientific projects on the environmental quality assessment.

**Funding:** This research is supported by the State order of Laboratory of Geoecology and Environmental Management of the Arctic of INEP KSC RAS No. 1021111018324-1 and the State order of Institute of Geology of Karelian Research Centre of RAS No. 1022040500826-4.

**Data Availability Statement:** The data that support the findings of this study are available on request from the corresponding author. The data are not publicly available due to privacy restrictions.

**Acknowledgments:** The author sincerely thank the colleagues V.A. Dauvalter for providing initial paleolimnological data and O.V. Petrova for creating of maps with the designation of lakes and key regions of paleolimnological studies.

**Conflicts of Interest:** There are no any competing interests as defined by Springer, or other interests that might be perceived to influence the results and/or discussion reported in this paper.

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
