# Peer review of "Geochemical Indicators for Paleolimnological Studies of the Anthropogenic Influence on the Environment of the Russian Federation: A Review"

_water, doi:10.3390/w15030420_

Round 1
Reviewer 1 Report
Author collected the published data and reviewed the elements concentration at three group areas in Russian, which may be helpful to understanding elements of lake sediment in Frigid Zone.
1) The abstract should provide the major finding. The present abstract showed too much detail. I suggested author shorted it and added comparing results among three types areas or other regions.
2) I suggest author provide a table or figure to introduce the major data source of references and number of references in the method section.
3) There are more datasets in three group areas. It may be better to use table or figure to show them at each group area.
4) If the figures (3,6,7,8,9,10,11,13-18) data cited from other references, it should be listed the references.
5) If there is one section to compare among three group areas or other regions, it will be better to know their variation or pollution level.
6) The length of review is too long, it should short it and provide the major conclusion.
Some areas should revise into Engilish, e.g. L280, unit in Figure 14.
Author Response
Reply to reviewer 01
Dear Reviewer,
I thank you for reviewing my publication. I have fixed the manuscript accordind part of your comments. The corrected text has already been sent to the editorial office. You can see our answers to your comments below.
The abstract should provide the major finding. The present abstract showed too much detail. I suggested author shorted it and added comparing results among three types areas or other regions.
I have changed abstract according your suggestion.
I suggest author provide a table or figure to introduce the major data source of references and number of references in the method section.
I think all you suggested is in section References. New tables or figures will increase the article that is large without it.
There are more datasets in three group areas. It may be better to use table or figure to show them at each group area.
I understand your suggestion, but I decided to leave everything unchanged in these sections. But I added brief conclutions after each subsections of Results and discusion.
If the figures (3,6,7,8,9,10,11,13-18) data cited from other references, it should be listed the references.
Everything has been changed.
If there is one section to compare among three group areas or other regions, it will be better to know their variation or pollution level.
I have added subsection "3.4. Comparison of element concentrations" with figure 19.
The length of review is too long, it should short it and provide the major conclusion.
I think such length is normal for review article. Besides, I don't know what section I can decrease.. For me, the length of each part of the article is quite appropriate, considering what a large collection of data I had to process.
Some areas should revise into Engilish, e.g. L280, unit in Figure 14.Abbreviations: Abbreviations need to be defined properly. Definitions for some of the abbreviations are missing.
Everything has been changed.
In the corrected text I also took into account the comments and suggestions of other reviewers.
With regards, author
Reviewer 2 Report
The author has carried out a well-defined review of pollution studies based on trace metals in the lake water bodies. The review is well designed barring a few spell checks and to be re-read and corrected for language presentation. grammatically. A few sentences have been corrected and this needs to be carried out for the entire manuscript. Figures, legends, and places and sites should be clear and the fonts to be increased.
Minor revision to be carried out. before accepting for publication.

Author Response
Reply to reviewer 02
Dear Reviewer,
I thank you for reviewing my publication. I have fixed the manuscript accordind all your comments. For instance, figures, legends, and places and sites were made more clear and the fonts were increased you suggested. The corrected text has already been sent to the editorial office.
In the corrected text I also took into account the comments and suggestions of other reviewers.
With regards, author
Reviewer 3 Report
I reviewed the manuscript. Various regions of Russia were reviewed in order to address pollution and its sources as well as the anthropogenic impact on environmental health and safety. The manuscript is interesting and can be published with only minor changes.
In my opinion, the author should provide a brief conclusion separate from the body of the text.
Author Response
Reply to reviewer 03
Dear Reviewer,
I thank you for reviewing my publication. I have fixed the manuscript accordind all your comments. For instance, I provide brief conclusions for each subsection in part of resuls and discusion. The corrected text has already been sent to the editorial office.
In the corrected text I also took into account the comments and suggestions of other reviewers.
With regards, author
Round 2
Reviewer 1 Report
Most of the comments have been considered by the author. I do not have further comments.